DOI: 10.1038/s41467-017-00479-7　　**OPEN**

# Editing an α-globin enhancer in primary human hematopoietic stem cells as a treatment for β-thalassemia

Sachith Mettananda [1,2], Chris A. Fisher[1], Deborah Hay[1], Mohsin Badat[1], Lynn Quek[1], Kevin Clark[3], Philip Hublitz[3], Damien Downes[1], Jon Kerry[1], Matthew Gosden[1], Jelena Telenius[1], Jackie A. Sloane-Stanley[1], Paula Faustino[4,5], Andreia Coelho[4], Jessica Doondeea[1], Batchimeg Usukhbayar[1], Paul Sopp[3], Jacqueline A. Sharpe[1], Jim R. Hughes [1], Paresh Vyas [1,6], Richard J. Gibbons[1] & Douglas R. Higgs [1,6]

β-Thalassemia is one of the most common inherited anemias, with no effective cure for most patients. The pathophysiology reflects an imbalance between α- and β-globin chains with an excess of free α-globin chains causing ineffective erythropoiesis and hemolysis. When α-thalassemia is co-inherited with β-thalassemia, excess free α-globin chains are reduced significantly ameliorating the clinical severity. Here we demonstrate the use of CRISPR/Cas9 genome editing of primary human hematopoietic stem/progenitor (CD34+) cells to emulate a natural mutation, which deletes the MCS-R2 α-globin enhancer and causes α-thalassemia. When edited CD34+ cells are differentiated into erythroid cells, we observe the expected reduction in α-globin expression and a correction of the pathologic globin chain imbalance in cells from patients with β-thalassemia. Xenograft assays show that a proportion of the edited CD34+ cells are long-term repopulating hematopoietic stem cells, demonstrating the potential of this approach for translation into a therapy for β-thalassemia.

[1] Medical Research Council (MRC) Molecular Hematology Unit, MRC Weatherall Institute of Molecular Medicine, University of Oxford, Oxford OX3 9DS, UK. [2] Department of Paediatrics, Faculty of Medicine, University of Kelaniya, Ragama 11010, Sri Lanka. [3] MRC Weatherall Institute of Molecular Medicine, University of Oxford, Oxford OX3 9DS, UK. [4] Human Genetics Department, National Institute of Health Dr. Ricardo Jorge, Av. Padre Cruz, Lisbon 1649-016, Portugal. [5] Institute of Environmental Health, Faculty of Medicine, University of Lisbon, Av. Prof. Egas Moniz, Lisbon 1649-028, Portugal. [6] Oxford National Institute for Health Research Biomedical Research Centre, Blood Theme, Oxford University Hospital, Oxford OX3 9DU, UK. Richard J. Gibbons and Douglas R. Higgs contributed equally to this work. Correspondence and requests for materials should be addressed to D.R.H. (email: doug.higgs@imm.ox.ac.uk)

Thalassemia is a disorder of hemoglobin synthesis characterized by severe anemia, which requires intensive supportive treatment from early childhood[1]. The most common and severe form of this disease (β-thalassemia) results from an absent or reduced production of normal β-globin chains[2]. Most aspects of the pathophysiology of β-thalassemia can be explained by the presence of excess α-globin chains, which can no longer pair with the reduced numbers of β-globin chains, found in patients with β-thalassemia, to produce normal hemoglobin tetramers ($\alpha_2\beta_2$). Excess α-globin chains precipitate both in red cell precursors (causing ineffective erythropoiesis) and mature red cells (causing hemolysis). Clinical and genetic data have clearly shown that when α-thalassemia is co-inherited with β-thalassemia, there is reduced expression of the α-globin genes, less globin chain imbalance, and reduced numbers of free α-globin chains, significantly ameliorating the clinical severity of β-thalassemia[3]. By contrast, inheritance of a higher than normal number of α-globin genes (5 or 6 rather than 4) substantially worsens the disease phenotype, emphasizing that excess α-globin chains are the major determinant of the clinical severity of β-thalassemia[4, 5].

The human α-globin gene locus is situated in the short arm of chromosome 16 with two copies of the α-globin gene on each chromosome (αα/αα). Expression of the α-globin genes is controlled by four enhancers (MCS-R1 to R4) located 10–50 kb upstream of the genes[6, 7]. Previous studies, including transgenic experiments combined with observations of naturally occurring mutations, have shown that a multi-species conserved sequence, which lies 40 kb upstream of the α-globin locus (MCS-R2, also known as HS-40), is the most powerful enhancer of α-globin gene expression[8–10]. We have previously characterized MCS-R2 in detail and shown that its activity is contained within a ~260 bp core fragment, including several well-conserved erythroid transcription factor binding sites[11]. A 1.1 kb deletion removing MCS-R2 in a humanized mouse model has been shown to result in a significant reduction of human α-globin expression[9]. More recently, Coelho et al.[12] reported a patient homozygous for a rare 3.3 kb deletion, which uniquely removes MCS-R2 and results in a significant downregulation of α-globin gene expression. Thus we hypothesized that if a targeted mutation of MCS-R2 was to be created it should result in a reduction of α-globin expression to levels beneficial to patients with β-thalassemia.

Genome editing using the CRISPR/Cas9 (clustered, regularly interspaced, short palindromic repeat/CRISPR-associated protein 9) system provides a realistic approach to the treatment of human genetic diseases including hemoglobinopathies[13]. These nucleases create double-strand breaks at specific, chosen locations in the genome and, when repaired, create mutations at the targeted sites[14]. Although CRISPR/Cas9 can be used to promote either homology-directed recombination (HDR) or non-homologous end joining, hematopoietic stem cells (HSC) are currently largely limited to the latter form of editing since, despite some recent developments[15], HDR remains an inefficient process in HSCs[16]. This means that deleting and inactivating an enhancer is currently a much more tractable approach to ameliorating β-thalassemia than directly repairing the β-globin gene or removing the α-globin genes by HDR.

Here we show the use of CRISPR/Cas9 genome editing technology to create a targeted mutation of the MCS-R2 core element in human HSCs to mimic the effects of natural mutations, which knockdown α-globin expression. We demonstrate successful knockdown of α-globin expression in vitro in erythroid cells generated by genome-edited HSCs to levels beneficial in β-thalassemia without perturbing erythroid differentiation or having detectable off-target events. Finally, we show that this form of engineering occurs in long-term repopulating HSC (LT-HSC), the cell population currently used in clinical practice

for HSC therapy in blood diseases, demonstrating the potential of this approach for translation into a therapy for β-thalassemia.

## Results

### A natural deletion that uniquely removes MCS-R2 enhancer.
A previously reported patient (MC) is homozygous for a very rare 3.3 kb deletion (called $(\alpha\alpha)^{ALT}$), which uniquely removes MCS-R2 but leaves the α-globin genes and all other enhancer elements intact[12]. MC is a member of a pedigree (Fig. 1a) that originates from Portugal and was evaluated in greater detail (Supplementary Table 1). Hematological analysis confirmed that MC has hypochromic microcytic anemia (Figs. 1b, c); 51% of his peripheral blood red blood cells (RBC) were positive for hemoglobin H (HbH) inclusion bodies confirming a diagnosis of α-thalassemia leading to HbH disease (Fig. 1d). α-Globin messenger RNA (mRNA) levels of MC's peripheral blood are reduced and multiplex ligation-dependent probe amplification analysis[12] confirmed the absence of MCS-R2 in its natural genomic location, but did not exclude the possibility of its presence in an ectopic locus. To examine this we performed Southern blot analysis using a probe specific to MCS-R2, which confirmed the absence of the MCS-R2 region anywhere in the genome (Fig. 1e).

Interestingly, in the $(\alpha\alpha)^{ALT}$ mutation, the 3.3 kb deletion is associated with the insertion of a 39 nucleotide orphan segment, whose sequence was not found anywhere in the human genome. Our search for canonical binding motifs for both general transcription factors, and erythroid-specific factors within this orphan sequence did not reveal motifs with any known regulatory potential. Next, we examined the presence of histone 4 acetylation (H4Ac) or binding of the stem cell leukemia (SCL) complex, which are signatures of active chromatin within this orphan insert sequence. Chromatin immunoprecipitation (ChIP-qPCR) across this sequence confirmed the absence of enrichment for these activating chromatin marks (Figs. 1f, g).

To assess whether the function of MCS-R2 was assumed by any other region within or beyond the α-globin cluster, we undertook ChIP-Seq (using an anti-pan-H4 acetylation antibody) to give the greatest likelihood of detecting any newly activated chromatin (Fig. 1h). No new putative regulatory elements were detected, with the only additional finding of note being the loss of a single H4Ac peak at the HBM promoter. This corresponds to a short deletion in the region of the HBM promoter, confirmed by Southern blot. This region does not contribute to normal α-globin transcription[17] and its loss is not associated with any expected reduction in α-globin levels.

These data support experimental studies in the mouse[18], which define MCS-R2 as the major (but not the only) regulatory element controlling expression of α-globin RNA. Its loss from the natural chromosomal environment results in a significant reduction of α-globin transcription in cis. The clinical phenotypes of thalassemia trait in the heterozygous patient RC and of HbH disease in patient MC, who carries this deletion in homozygosity, demonstrate that deletion of MCS-R2 causes a significant (>50%), but not total reduction in α-globin expression in both homozygous and heterozygous states. However, of importance for this study, the absence of MCS-R2 in the homozygous state did not result in any other phenotypic abnormalities in MC, demonstrating that its unique functional significance is confined to expression of the α-globin locus.

### Targeted in vitro deletion of MCS-R2 to knockdown α-globin.
Next, we hypothesized that creating a targeted deletion of MCS-R2 would phenocopy the effects seen in MC and RC in homozygous and heterozygous states, respectively. Thus the removal of the ~260 bp core element of MCS-R2 in human

erythroid cells should result in a reduction in α-globin expression to levels previously shown to be beneficial in patients with β-thalassemia. To do this, we designed several short guide RNAs (sgRNA) complementary to target sequences both up- and downstream of the MCS-R2 core element (Fig. 2a; Supplementary Table 2) to generate CRISPR plasmid pairs to use in combination to create two double-strand breaks at either end of the enhancer. When repaired this should introduce targeted deletions. These sgRNAs were cloned into a plasmid vector (Addgene plasmid 48138: pSpCas9(BB)-2A-GFP (pX458)) that contains a gene

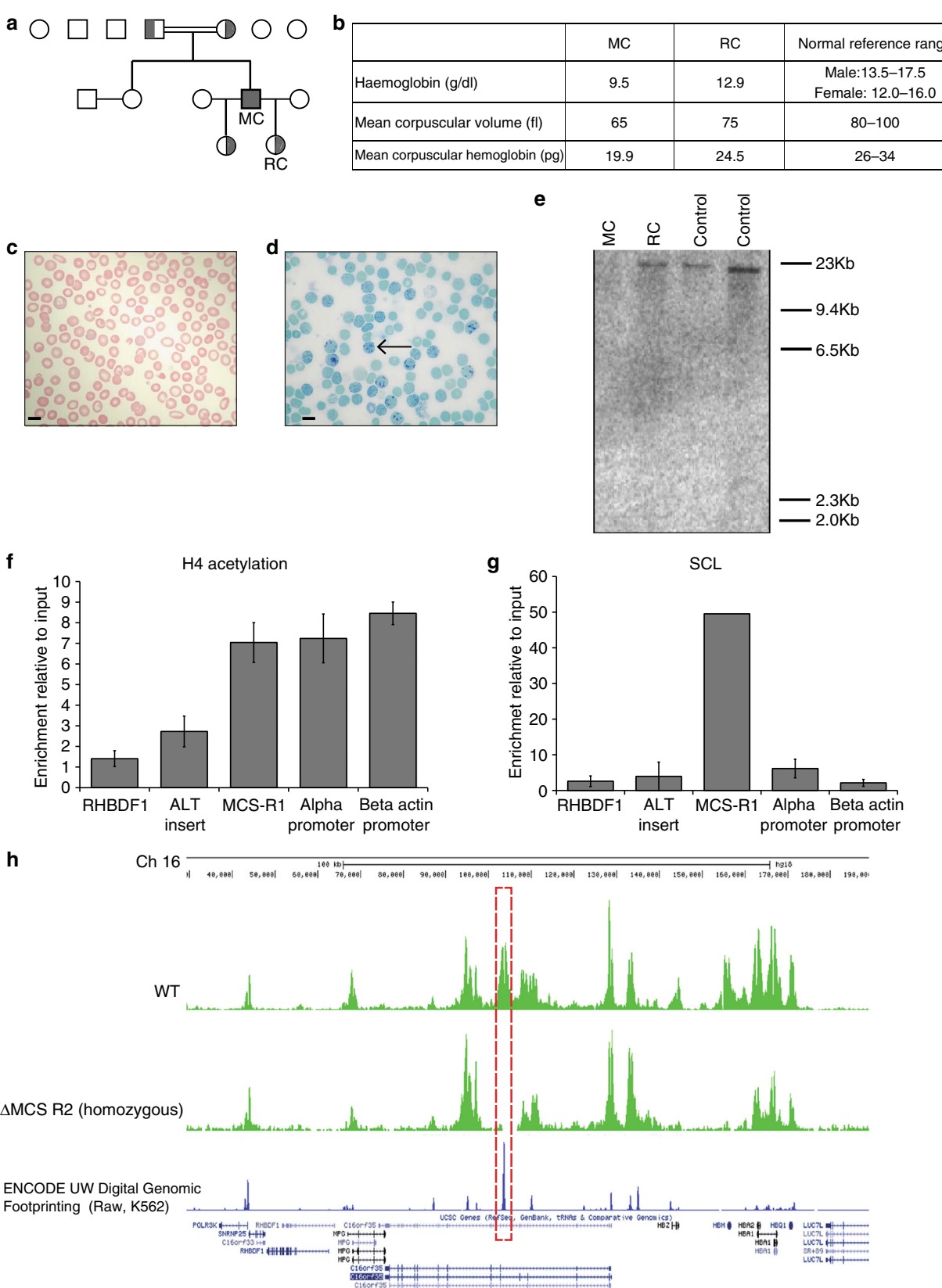

encoding green fluorescent protein (GFP) fused to a Cas9 gene. We then co-transfected primary human CD34+ cells, obtained from normal, non-thalassemia controls, with pairs of CRISPR plasmids (Supplementary Fig. 1). The live-transfected cell populations were sorted by GFP expression (Supplementary Fig. 2; Fig. 2b) and screened for the predicted mutations. Through this preliminary screening we identified four optimal CRISPR plasmid pairs, which generated targeted deletions at the highest frequencies (Supplementary Fig. 3). Deletion efficiency of these CRISPR pairs was verified by two independent methods and was consistently over 70% when using three out of four pairs (Figs. 2c, d). The accuracy of editing was then determined by Sanger sequencing (Fig. 2e). Next we differentiated these transfected CD34+ cells along the erythroid lineage to evaluate globin gene expression. As expected, and similar to the observations made in MC, with all four CRISPR plasmid pairs, deletions of MCS-R2 (ΔMCS-R2) resulted in selective reduction in α-globin expression without altering the expression of the β-globin gene (Fig. 2f).

**Single-cell assays confirm α-globin knockdown by ΔMCS-R2.** Next, to precisely dissect the effects of CRISPR/Cas9-mediated ΔMCS-R2, we performed single-cell assays using Terasaki multiwell plates, which were superior to the traditional methylcellulose-based colony assays in minimizing cross-contamination between clones. We transfected CD34+ cells with a pair of CRISPR plasmids (Cr2 + Cr12) and then sorted these cells using flow cytometry to generate single cells in individual wells. Following erythroid differentiation, individual clones were assayed for genotype and globin gene expression, which enabled us to determine the correlations between the dosages of mutant alleles and globin gene expression, and confirm the editing efficiency. Genotype analysis of these clones by PCR and sequencing revealed a variety of editing outcomes not only limited to the expected mutations resulting in a mixture of different genotypes, which included MCS-R2 non-deletions (wild type), heterozygous and homozygous ΔMCS-R2 mutations, MCS-R2 inversions, and unpredicted mutations (Figs. 3a, b; Supplementary Fig. 4). Analysis of globin gene expression of individual erythroid clones demonstrated significant reductions in α/β-globin mRNA ratios in clones with heterozygous (median 36% compared to normal) and homozygous (median 3% compared to normal) ΔMCS-R2 mutations (Fig. 3c) thus confirming the selective downregulation of α-globin by deletion of MCS-R2. The α/β-globin mRNA ratios in clones with inversions of the MCS-R2 were similar to those without mutations (Fig. 3d). Similar results were observed in single-cell clone analysis of CD34+ cells transfected with another independent CRISPR pair (Cr1 + Cr8) thus corroborating the results (Supplementary Figs. 5 and 6).

**Genome-edited cells have no detectable off-target effects.** To explore any off-target effects of MCS-R2 deletion by CRISPR, we identified potential genomic off-target loci for the four sgRNA (Cr1, Cr2, Cr8, and Cr12) using the Sanger off-target prediction tool (Supplementary Table 3). Sequencing of the DNA from genome-edited clones did not identify any off-target activity at these sites (Fig. 3e; Supplementary Fig. 5; Supplementary Table 4) confirming the specific on-target activity of the CRISPR plasmids used here.

To examine the effect of deleting MCS-R2 on the differentiation of CD34+ cells following genome editing, we cultured the edited cells in conditions that favor the erythroid lineage. Morphological analysis of stained cytospins and immunophenotypic characterization using erythroid-specific cell surface markers were performed. CRISPR-edited cells demonstrated the same patterns of differentiation as unedited control cells (Supplementary Fig. 7).

**Genome-edited HSCs retain long-term repopulating ability.** If genome editing of MCS-R2 is to become a realistic therapeutic approach in β-thalassemia, targeted mutations must be produced in LT-HSC. Human CD34+ cells are used in all current stem cell transplantation protocols for blood diseases, but they are known to be a heterogeneous collection of hematopoietic progenitors and rare LT-HSCs (so called HSCs;<0.5% of total nucleated bone marrow cells). To exclude the possibility that genome editing occurred in progenitors but not LT-HSCs, CRISPR-transfected CD34+ cells were injected into the bone marrows of four sub-lethally irradiated female NSG mice (80,000 cells per mouse) and their bone marrow was harvested after 12 weeks (Figs. 4a, b). All four mice demonstrated long-term multi-lineage (lymphoid and myeloid) engraftment (range 0.8–34% human leukocytes), and three had particularly high levels (27–34% hCD45+) (Fig. 4c; Supplementary Fig. 8). Human CD45+ cells obtained from these bone marrows were sorted and DNA was screened for the predicted edits from these three mice: two had genome-edited cells of which one (xenograft #3) had an edited allele frequency of 71% (Fig. 4d). Although not found in the in vitro analyses, in this analysis we found a slightly larger deletion (271 bp instead of 241 bp) presumably due to exonuclease cut back from the CRISPR/Cas9 cut sites (Fig. 4e). Nevertheless, this deletion removes the key transcription factor binding sites as in the experimental plan. Flow cytometry analysis of harvested bone marrow of xenograft #3 mice showed that 15.5% of all human CD45+ cells were positive for human CD34+.

To determine if we had successfully edited LT-HSC, we performed serial, secondary transplantation with hCD45+ cells from the primary engrafted mice (100,000 cells per mouse, n = 3). We detected multi-lineage engraftment of hCD45+ cells in 2/3 mice (range 29.7–36.9% human leukocytes). We also performed

**Fig. 1** Characterization of a rare natural mutation ((αα)$^{ALT}$) confined to MCS-R2. **a** Kindred showing patient (MC) homozygous for the (αα)$^{ALT}$ mutation and his heterozygote daughters. **b** Hemoglobin level and red cell parameters of MC and RC (who is heterozygous for the (αα)$^{ALT}$ mutation). **c** Peripheral blood smear, Giemsa stained, from patient MC, which shows anisocytosis and poikilocytosis with some irregularly contracted cells; *scale bar* represents 10 μm. **d** Peripheral blood Hemoglobin H (*HbH*) preparation, brilliant cresyl blue stained, from patient MC, which shows HbH inclusions; *scale bar* represents 10 μm. **e** Southern blot analysis using a probe specific for the core of MCS-R2, which gives an 19 kb band on BgIII digest genomic DNA from normal controls and RC but not from MC confirming the absence of this segment. **f, g** Analysis of enrichment of histone 4 (H4) acetylation **f** and SCL **g** by ChIP-qPCR in in vitro differentiated primary erythroid cells (Fibach culture[35]) of MC harvested at intermediated erythroblasts stage (day 8–10 of Phase 2). The *y*-axis represents enrichment relative to ChIP input, normalized to 18S control region. Mean values are presented and *error bars* represent SD (n = 3). Amplicons are labeled thus: RHBDF1, intronic amplicon within the Rhomboid gene *RHBDF1*, used as a negative control; ALT insert, amplicon specific for 39 bp insert; MCS-R1, the human MCS-R1 which is one of the other enhancers of α-globin; alpha-promoter, α-globin promoter; beta actin promoter, a control amplicon over the β-actin promoter. **h** Analysis of the abundance of panH4 acetylation across the α-globin locus by ChIP-seq in in vitro differentiated primary erythroid cells harvested at intermediated erythroblasts stage from patient MC (*lower track*) and a normal control (*upper track*), showing the absence of the peak over MCS-R2 in patient MC (*red dashed box*)

methylcellulose colony assays with CD34+ cells harvested from the secondary xenograft, which again demonstrated the presence of granulocyte/macrophage and erythroid progenitor activity (Fig. 4f). Sanger sequencing of 40 colonies (20 erythroid and 20 myeloid) confirmed deletion of MCS-R2. These data confirm that we have successfully edited LT-HSC, which are able to reconstitute myeloid, lymphoid, and erythroid hematopoiesis.

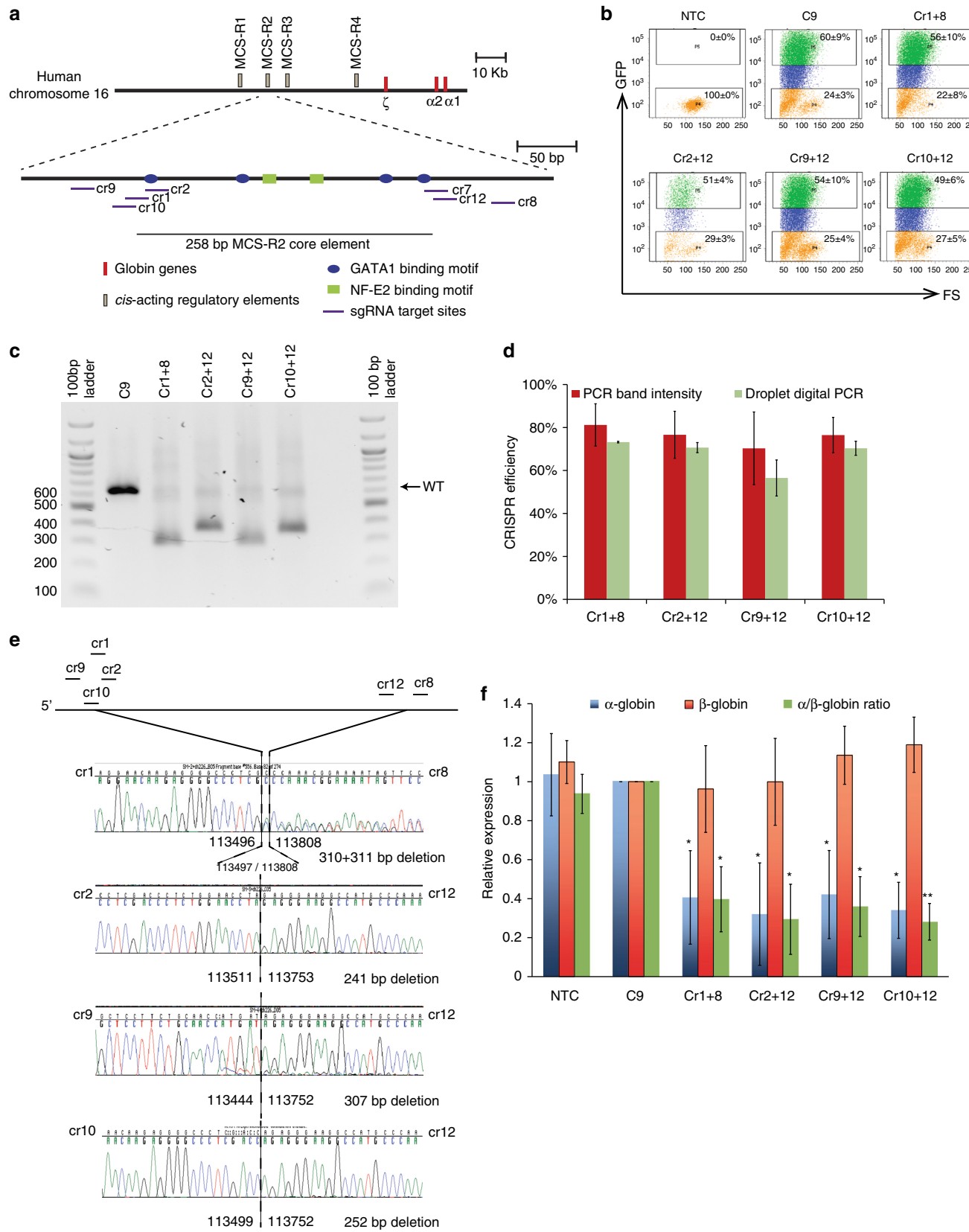

**ΔMCS-R2 restores globin balance in β-thalassemia cells.**
Finally, to examine the ability of the edited deletion of MCS-R2 to rectify globin chain imbalance in the erythroid cells of patients with β-thalassemia, we performed single colony analysis on CRISPR-edited CD34+ cells obtained from patients with HbE β-thalassemia (Supplementary Table 5). Genome editing of CD34+ cells obtained from patients with HbE β-thalassemia produced a mixture of different genotypes similar to those observed in normal controls with comparable mutation frequencies (Supplementary Fig. 9; Fig. 5a). As expected, the α/β globin mRNA ratios in unmodified erythroid cells from patients with HbE β-thalassemia were higher than α/β globin mRNA ratios in unmodified, normal erythroid cells. However when a heterozygous ΔMCS-R2 mutation was edited into the HbE β-thalassemia cells, the α/β globin mRNA ratio decreased and many individual clones had α/β globin mRNA ratios similar to normal control clones (Fig. 5b). When both alleles at MCS-R2 were edited in cells derived from patients with HbE β-thalassemia, the α/β globin ratios were rebalanced and many individual clones had α/β globin mRNA ratios similar to those found in normal control cells (Fig. 5b). Analysis of α/all β-like globins ($\alpha/(\alpha+\gamma)$) ratios demonstrated similar trends (Fig. 5c). These observations confirm that editing of MCS-R2 can efficiently restore globin chain balance in erythroid cells in patients with β-thalassemia and we predict that this would result in phenotypic improvements and improved survival of genome-edited cells with reduced levels of erythroid cell destruction and anemia in vivo.

## Discussion

The clinical management of β-thalassemia still largely depends on supportive treatment with RBC transfusions and iron chelation in the majority of patients[1]. Allogenic bone marrow transplantation remains the only curative treatment; however its usefulness is limited to a minority of patients who have HLA-matched sibling donors[19]. Several new therapies for β-thalassemia are currently being investigated[20]. Except for a few studies[21] all of these new and experimental therapies aim to resolve globin chain imbalance by increasing the production of γ-globin and fetal hemoglobin by genome editing of transcription factors[22, 23] (eg: *BCL11A*[24] and *LRF/ZBTB7A*[25]) or by pharmacological methods (eg: histone deacetylase inhibitors[26] and lysine-specific demethylase 1[27]). Alternatively, here we present a novel approach by directly reducing expression of α-globin. Considering the central role of excess α-globin chains in the pathophysiology of β-thalassemia, and the extensive clinical evidence showing how co-inheritance of α-thalassemia ameliorates β-thalassemia, this presents an extremely promising approach to curing this disease until efficient gene correction of the defective β-globin by HDR in stem cells becomes routinely possible. This approach can be used on its own or to complement other on-going efforts of increasing the production of fetal hemoglobin.

We have previously summarized the findings from several clinical studies showing that a reduction of α-globin expression to 75–25% of normal is an effective, safe, and tolerable level to provide sustainable beneficial effects in patients with β-thalassemia[3]. In genome-edited normal control cells many of the clones with heterozygous or homozygous deletions of MCS-R2 demonstrated reduction of the α/β ratio to this desirable range. In cells with HbE β-thalassemia mutations, targeted deletion of MCS-R2 in both heterozygous and homozygous states resulted in amelioration of the disturbed globin ratios.

In this paper we have demonstrated the use of CRISPR/Cas9 system for genome editing of MCS-R2, the major α-globin enhancer, in human primary LT-HSCs. CD34+ cells, including LT-HSCs, are readily obtainable at a clinical scale and are currently used in all stem cell transplantation protocols for blood diseases. Thus the hematopoietic system offers the ideal model for translation of genome editing for clinical benefit, and successful genome editing of lymphoid cells using a zinc-finger nuclease-based technique has already progressed to clinical trials[28]. Translation of our strategy into clinical application would require a harvest of CD34+ cells from patients with β-thalassemia, ex vivo genome editing, and an autograft. This would therefore avoid the potentially life-threatening complications of graft rejection and graft versus host disease associated with allografting for patients with hemoglobinopathies.

Using our protocol, we currently achieve transfection efficiency as high as 75% and in transfected cells the editing efficiency was 70–80%, providing an overall editing rate of 50–60% of live CD34+ cells. Previously, it has been shown that chimerism levels of only 10–20% of normal HSCs are sufficient to result in nearly complete hematologic and pathologic correction of β-thalassemia[29, 30]. Therefore, using our strategy, we should be able to provide sufficient numbers of genome-edited HSCs to produce a clinically significant beneficial effect in patients with β-thalassemia.

One potential limitation of this strategy is that deletion of MCS-R2 could lead to profound reduction in α-globin expression thus causing a critical decrease in total globin synthesis in some edited cells. However, it is likely that erythroid cells with more balanced globin chain synthesis generated from genome-edited HSCs will have less ineffective erythropoiesis and hemolysis and remain in the circulation for the full life-span thus demonstrating selective advantage for survival in vivo. Furthermore, genome engineering of the MCS-R2 enhancer could be used to fine-tune the level of α-globin expression by limiting the edit to already identified transcription factor binding sites within the enhancer. Therefore, the issue of excessive downregulation of α-globin need not be a significant barrier during future clinical use.

Xenograft assays performed here show that CRISPR/Cas9 genome editing is not limited to progenitor cells but also occurs in LT-HSCs. These are the cells, which must be edited if a sustainable improvement is to be achieved in patients with β-thalassemia. We have shown that edited HSCs give rise to multiple lineages of hematopoietic cells in vivo. In the xenograft

**Fig. 2** Deletion of MCS-R2 using CRISPR/Cas9 genome editing in human CD34+ cells. **a** Schematic of sgRNA target sites. Four sgRNAs (Cr1, Cr2, Cr9, and Cr10) were designed to target the 5′ end of the MCS-R2 core element, whereas three sgRNAs (Cr7, Cr8, and Cr12) target the 3′ end. **b** Representative flow cytometry plots showing GFP expression and forward scatter (FS) after gating for live cells in non-transfection control (NTC), Cas9 control (C9), and CRISPR/Cas9 plasmid pair-transfected cells. *Orange*: GFP negative, *blue*: GFP positive (low), and *green*: GFP positive (high). Mean and SD of the percentage of cells within each region is indicated ($n=3$). Gating strategy is shown in Supplementary Fig. 10a. **c** Representative gel electrophoresis image of genomic DNA extracted from cells targeted by four CRISPR/Cas9 plasmid pairs analyzed by PCR. **d** Gene editing deletion induction efficiency as measured independently by percentages of mutated alleles determined by band size in end-point PCR and subsequent Sanger sequence analysis and by determining inverse of proportions of amplicons inside: outside deletion (amplicons of same length) by multiplexed droplet digital PCR; mean values are presented and *error bars* represent SD ($n=3$). **e** Characterization of deletion break points by sequencing (co-ordinates from Hum Mar 2006 (NCBI36/hg18) assembly). **f** α-and β-globin gene expression normalized to the expression of *RPL13A* and α/β-globin mRNA ratios relative to Cas9 control (C9) analyzed by qPCR; *error bars* represent SD ($n=3$); *$P < 0.05$ and **$P < 0.01$ relative to C9 (Student's *t*-test). C9 Cas9-only control

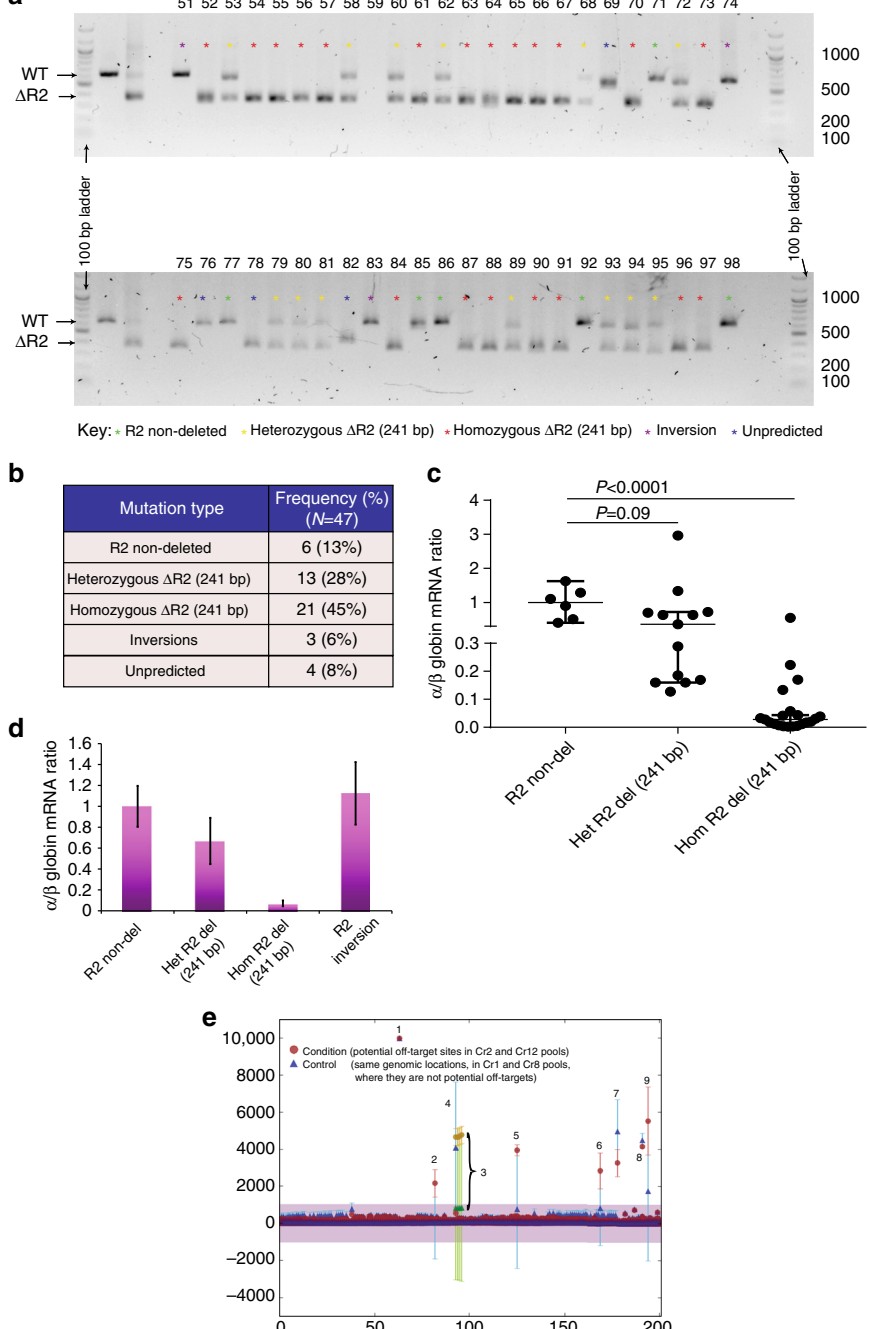

**Fig. 3** Single-cell clone analysis of targeted deletion of MCS-R2. **a** Gel electrophoresis image of genomic DNA from 48 individual single-cell clones genome-edited using CRISPR pair Cr2 and Cr12 (from three biological independent donors) analyzed by PCR. The amplicon from the wild-type allele is 613 bp and the mutated amplicon is 372 bp. Clones are numbered 51–98 and clone 59 failed to amplify. Extended genotype analysis of these clones by sequencing is presented in Supplementary Fig. 4. **b** Frequency of different types of mutations generated. **c** $\alpha/\beta$-globin mRNA ratios of individual clones of erythroid cells which are non-deleted (R2 non-del) ($n = 6$) and heterozygous (Het R2 del.) ($n = 13$) or homozygous (Hom R2 del.) ($n = 21$) for a 241 bp deletion of MCS-R2 region analyzed by qPCR; median (*horizontal bar*) and 95% confidence interval (*error bar*) are shown and P-values were calculated using Mann–Whitney U-test. **d** $\alpha/\beta$-globin mRNA ratios of erythroid cells, which has no deletion ($n = 6$), heterozygous deletion ($n = 13$), homozygous deletion ($n = 21$), and inversion ($n = 3$) of MCS-R2 analyzed by qPCR. Means and SEM are shown. **e** Meta-plot of all off-target loci for CRISPR pair Cr2 and Cr12. All captured sites are plotted on the same x-axis, showing $\pm 100$ bases from each potential off-target site. Counts deviating from the reference sequence, which are normalized to 10,000 counts are plotted in the y-axis. The values are means of the libraries where each library is a pool of five independent clones. The shaded *violet* area denotes $\pm 1000$ counts and only data over this threshold were considered as off-target. *Error bars* represent SEM for each base at each locus. Potential off-target hits for Cr2 and Cr12 (condition) are plotted alongside those of a control group (control). The numbers are annotated in Supplementary Table 4. All of the variations from the reference sequence were shown to be known SNPs or indels or novel variations common to both control and condition and therefore unrelated to potential off-target activity

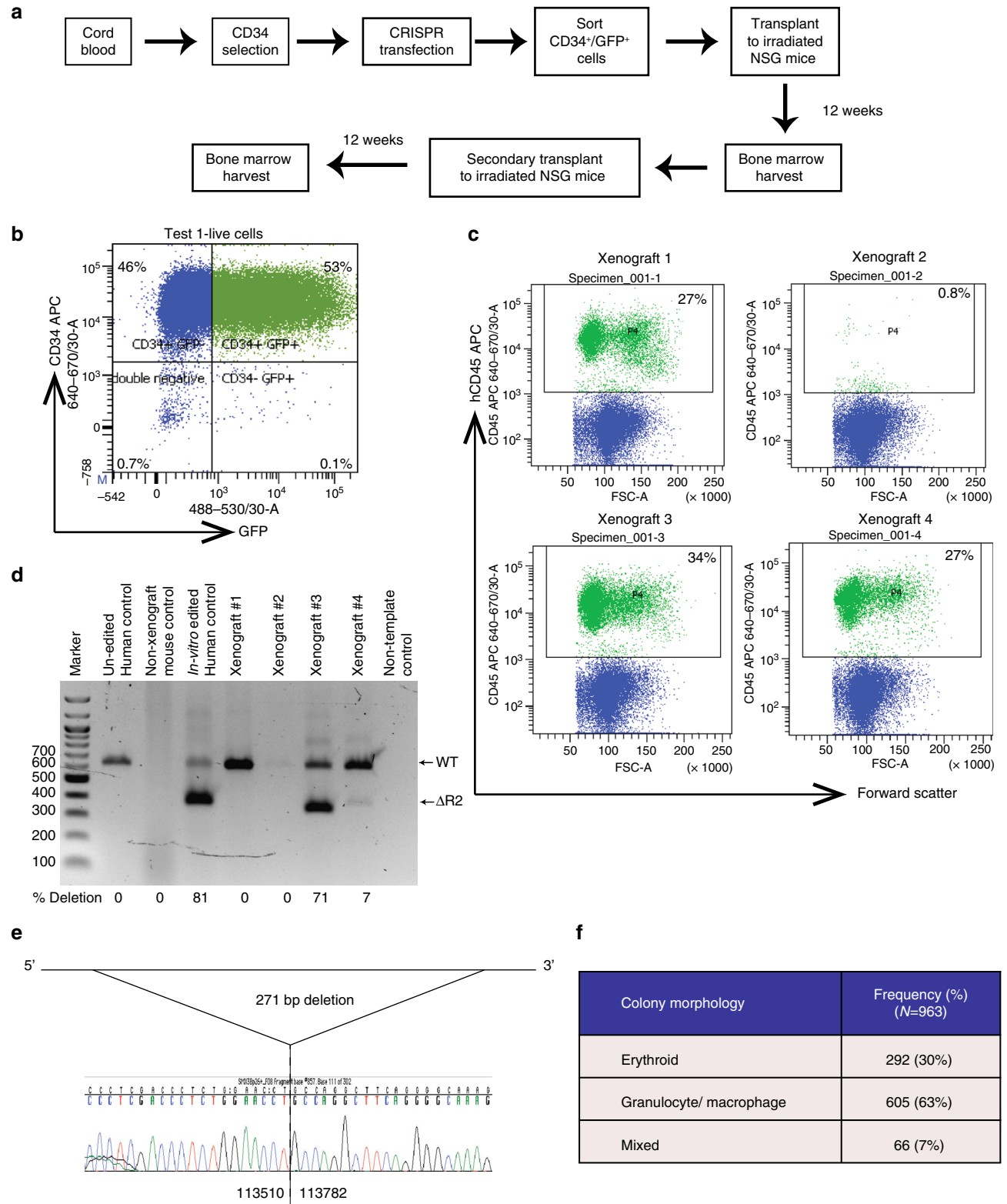

**Fig. 4** Xenograft assay. **a** Schematic of the workflow of mouse xenograft experiment. **b** Flow cytometry analysis of cells following CRISPR transfection demonstrating transfected (positive for GFP) CD34+ HSPCs. Gating strategy is shown in Supplementary Fig. 10b. **c** Flow cytometry plots of harvested bone marrow from xenograft mice gated for live cells demonstrating hCD45 expression. Gating strategy is shown in Supplementary Fig. 10c. **d** Gel electrophoresis image of genomic DNA extracted from human cells obtained from xenograft mouse analyzed by PCR demonstrating genome-edited bands. **e** Characterization of deletion break points by sequencing (co-ordinates from Hum Mar 2006 (NCBI36/hg18) Assembly) **f** Morphological analysis of hematopoietic colonies grown in methyl cellulose which were generated by HSCs harvested from secondary transplant mice (two independent mice)

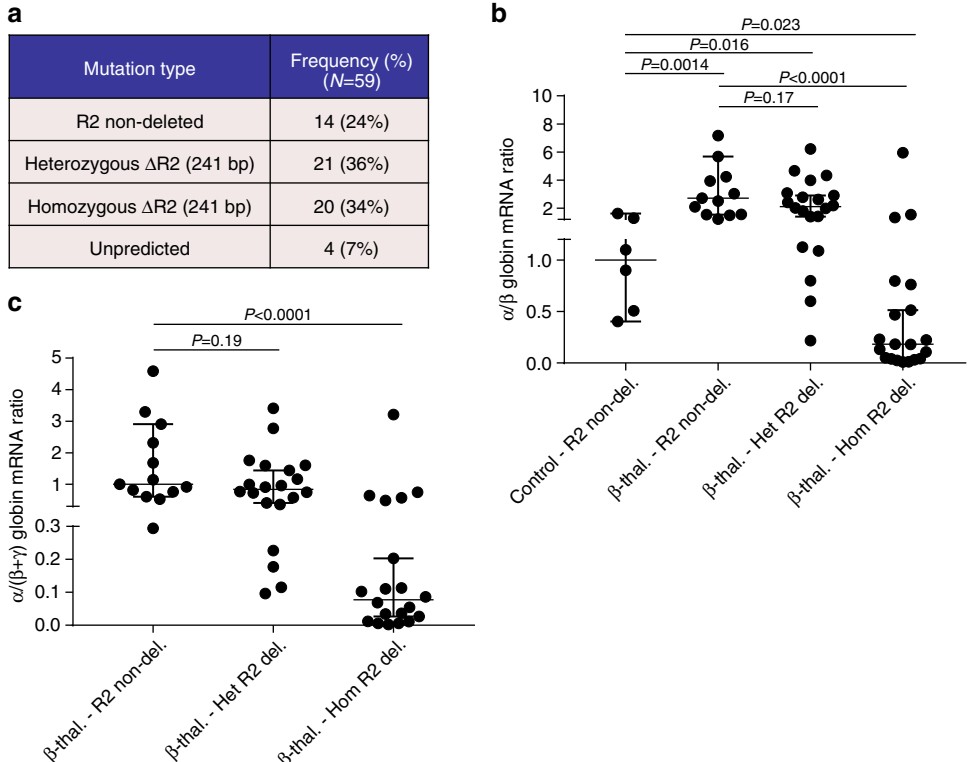

**Fig. 5** Deletion of MCS-R2 in CD34+ cells from HbE β-thalassemia patients using Cr2 + Cr12. **a** Frequency of different types of mutations generated. **b** α/β-globin mRNA ratios of individual clones of erythroid cells which are non-deleted normal control (n = 6), non-deleted HbE β-thalassemia (n = 13), HbE β-thalassemia heterozygous for the deletion of MCS-R2 (n = 20), and HbE β-thalassemia homozygous for the deletion of MCS-R2 (n = 20) analyzed by qPCR; median (*horizontal bar*) and 95% confidence interval (*error bar*) are shown and P-values were calculated using Mann–Whitney U-test. **c** Data for α/(β + γ) globin mRNA ratios for individual clones of HbE β-thalassemia erythroid cells grouped according to genotype and normalized to median of non-deleted HbE β-thalassemia clones; median (*horizontal bar*) and 95% confidence interval (*error bar*) are shown and P-values were calculated using Mann–Whitney U-test

assays we observed a single, slightly larger deletion in mice that had genome-edited cells. This suggests highly oligo or monoclonal reconstitution that may have resulted from a limited number or even a single LT-HSC. This is not entirely surprising as LT-HSCs are rare within the CD34+ population of cells and this does not detract from the conclusion that the editing procedure does result in the engineering of this population of cells. A xenografted mouse is not a suitable model in which to study erythropoiesis from transplanted cells and therefore we were not able to analyze the effect of ΔMCS-R2 on globin gene expression in vivo. However, there is no reason to think that expression in vivo will be any different from that seen in erythroid cells differentiated from CD34+ cells in culture.

An important consideration is whether editing a major enhancer would have effects on any other features of erythropoiesis or in other tissues. We addressed this by detailed analysis of a family with a rare natural deletion of MCS-R2. The two affected individuals with heterozygous and homozygous mutations of MCS-R2 are healthy with no other obvious phenotypes except for anemia validating MCS-R2 as a safe region of the genome to be targeted by genome editing. Furthermore, detailed evaluation of predicted off-target effects of genome editing tools used in our study did not reveal any off-target activity and the effects were specific to the target loci.

In conclusion, we have shown that it is feasible to edit LT-HSCs and to inactivate an enhancer, which specifically and uniquely regulates α-globin expression. Importantly, this reduces α-globin expression to the levels known to be required to achieve beneficial effects in patients with β-thalassemia. These findings

open up a new approach to ameliorate this life-limiting disease and provide useful insights into plausible therapeutic approaches to human genetic diseases in general.

## Methods

**Cell culture**. Human umbilical cord blood and adult peripheral blood buffy coat residues were purchased from the National Health Service (NHS) Blood and Transplant, UK. Samples from β-thalassemia patients were collected after obtaining informed written consent. Ethical approval for the study was granted by North West Research Ethics Committee of NHS National Research Ethics Services (reference no. 03/08/097). Mononuclear cells were separated using Histopaque-1077 Hybri-Max (Sigma), the CD34+ cells were purified using a CD34 MicroBead Kit (Miltenyi Biotech Cat. 130-046-702) and purity confirmed by FACS. Primary CD34+ cells were not tested for mycoplasma contamination. Then the CD34+ cells were differentiated into erythroid cells over 21 days using a two-phase liquid culture system which used StemSpan SFEM II (Stemcell Technologies). Phase 1 medium was supplemented with stem cell factor, interleukin-3, human recombinant erythropoietin (EPO) (0.5 U/ml) and cholesterol-rich lipids. Phase 2 medium was similar to phase 1 medium except for addition of iron saturated holotransferrin and higher concentration of EPO (3 U/ml). In single-cell experiments, single cells were sorted into the wells in Terasaki multiwell plates and were cultured in 20 μl culture medium.

**CRISPR plasmids**. Several sgRNA-targeting MCS-R2 were cloned into the Addgene plasmid 48138 (pSpCas9(BB)-2A-GFP (pX458)) backbone vector. Guides were cloned into the BbsI restriction site containing two Guanine residues at positions 0 and 1 of the protospacer, the sequence of the targeting guides are listed in Supplementary Table 2.

**Transfection of CD34+ cells**. Transfection of CRISPR plasmid DNA into CD34+ cells was done using Amaxa human CD34+ cell nucleofector kit (Lonza) and Amaxa nucleofector I device (Lonza)[31]. Culture medium containing $3 \times 10^5$–$1 \times 10^6$ cells was spun at 1200 rpm for 5 min. The cell pellet was

resuspended in 100 µl nucleofection solution and was nucleofected with 10 µg CRISPR plasmid DNA in the cuvette provided using the U-08 program. Cell suspension was incubated at room temperature for 10 min in nucleofection solution before transferring it to antibiotic-free medium.

**Flow cytometry**. Antibody labeling of washed cells was done using the following anti-human antibodies; allophycocyanin (APC)-conjugated anti-CD34, fluorescein isothiocyanate (FITC)-conjugated anti-CD71, phycoerythrin (PE) conjugated anti-CD235a, APC-conjugated anti-CD45, FITC-conjugated anti-CD19, and PE-conjugated anti-CD33. Dead cells were identified by Hoechst 33258 pentahydrate nucleic acid stain (Invitrogen) and were excluded. Analysis was performed on Cyan ADP (Beckman Coulter) analyzer using Summit v4.3 and FlowJo V10 softwares and cell sorting was performed in a BD FACSAria Fusion cell sorter (BD Biosciences). Details of flow cytometry gating strategy are given in Supplementary Fig. 10.

**Antibodies used for flow cytometry**. Antibodies to the following proteins were used: APC-conjugated anti-CD34 (Miltenyl Biotec Cat. 130-090-654; dilution 1:100), FITC-conjugated anti-CD71 (BD Biosciences Cat. 555536; dilution 1:100), PE-conjugated anti-CD235a (BD Biosciences Cat. 340947; dilution 1:500), APC-conjugated anti-CD45 (BioLegend Cat. 304012 Clone HI30; dilution 1:100), FITC-conjugated anti-CD19 (BioLegend Cat. 302206 Clone HIB19; dilution 1:100), and PE-conjugated anti-CD33 (BioLegend Cat. 303404 Clone WM53; dilution 1:100). Dead cells were identified by Hoechst 33258 pentahydrate nucleic acid stain (Invitrogen; dilution 1:10,000) and were excluded.

**DNA extraction and PCR**. Genomic DNA from cell pellets was extracted using DNeasy blood and tissue kit (Qiagen) and genomic DNA from small cell numbers was amplified directly from cell lysate using Illustra single-cell GenomiPhi DNA amplification kit (GE Healthcare). PCR reactions across MCS-R2 region were performed with AmpliTaq Gold 360 PCR master mix (Invitrogen) using a custom designed primer pair (forward 5′-TGGTCCTGAAGGATGAGAAG-3′; reverse 5′-AGCAACAGTCCTTTCTCTGG-3′). Uncropped scans of all PCR gels are shown in the Supplementary Fig. 11.

**Droplet digital PCR**. About 20 ng genomic DNA was amplified with a primer pair, one set with a primer within the deletion (5′-GGCCCAGTTATCTGCTC CCTCAAGT-3′) and a primer upstream (5′-AGGCCCATATCTCTGCCCAA-GAGC-3′) and the other set both 3.9 kb downstream (forward 5′-GGGGACTT TTGCCATGCCTGAAGTAGA-3′; reverse 5′-GGCCCCACTCCCTGATCT-TAACCATTT-3′), using the QX200 Droplet Digital PCR System. Around 20,000 droplets were evaluated and the gene editing efficiency of the different CRISPR sgRNA pairs was estimated by comparing the two primer sets against a negative control (vector only).

**Identification of potential off-target loci for CRISPR plasmids**. Potential genic off-target loci were identified using the Sanger off-target prediction tool (http://www.sanger.ac.uk/htgt/wge/find_off_targets_by_seq_allowing_up_to_four_mismatches) (Supplementary Table 3). These loci were captured by hybridization to complementary 50 bp biotinylated oligos prior to sequencing; these oligonucleotides (IDT) were designed to bind within 50 bp of either side of the CRISPR/Cas9 target and off-target sites using an in-house PERL script (Off-TargetProbes.pl). For each site, one oligo from each adjacent window was selected; selected oligos had the lowest BLAT density score and were only used if the score was below 35 and no simple repeats were present[32]. Chromosome capture oligos were successfully designed for 53/57 potential off-target sites: 22 potential off-target sites for Cr1, 6 potential off-target sites for Cr8, 8 potential off-target sites for Cr2, and 17 potential off-target sites for Cr12. (details in the Supplementary Table 3). Most sites had two capture oligos, but for some it was possible to design only one.

**Analysis of off-target loci**. Whole-genome amplified DNA was prepared (Illustra single-cell GenomiPhi DNA amplification kit, GE Healthcare) from 30 clones and combined in pools of five clones, each contributing equal amounts of DNA, and sonicated to 200 bp (Covaris S220 sonicator). Sonicated DNA was purified with Agencourt AMPure XP beads (Beckman Coulter) and indexed using NebNext Ultra II (New England Biolabs). Fragment size was confirmed before and after indexing using the D1000 Tapestation (Agilent).

The capture was performed using four pools of five clones for Cr2 and Cr12, and for two pools of five clones for Cr1 and Cr8. Up to 2 µg of each library pool was combined for multiplexed capture with capture-based enrichment of target and off-target regions. This was carried out in two successive rounds of hybridization, biotin pull down, and amplification with SeqCap EZ (Nimblegen) and M-280 Streptavadin Dynabeads (ThermoFisher) following the NG Capture-C method[33]. Each of these pools was sequenced as a separate library with the captured fragments sequenced in a 150 bp paired end (300 cycles mid output) run on the MiSeq Platform (Illumina).

The reads were mapped to the hg38 genome build, using bowtie2, (default parameters, —maxins 700), allowing only paired concordant mapping to be reported. The un-mapping read pairs were trimmed with trim_galore

(default parameters), and re-mapped in bowtie2. This resulted in 91–93% mapping. STAR mapping was used to map the remaining reads (first end-to-end mapping, then setting insertion elongation penalty to zero, and finally also allowing single end mapping). Each of these rounds mapped ~90% of the previously unmapped reads, resulting in overall mapping of 99.9%. No new sequencing artifacts were introduced by this process. The mapped data were compared to the reference sequence, by using samtools mpileup, and further excluding all bases in reads with lower quality score than 30, by using varscan. This resulted in read depth for each base of 40,000–200,000 (except sample 4 for Cr2 and 12, which had 13,000). The counts were normalized to 10,000 counts.

**RNA extraction and qRT-PCR**. Total RNA was purified using RNeasy mini kit (Qiagen) and cDNA from small number of cells was directly prepared from cell lysate using TaqMan gene expression cells-to-CT kit (Life Technologies). All qRT-PCR reactions were performed using inventoried TaqMan assays (Applied Biosystems; TaqMan IDs: HBA2/HBA1-Hs00361191_g1, HBB-Hs00747223_g1, HBG-Hs00361131_g1 and RPL13A-Hs03043885_g1) in technical triplicate in 7500 fast real-time PCR system (Applied Biosystems) according to the manufacturer's protocol. Data were analyzed by 7500 software v2.0.6 using the delta delta CT method.

**ChIP and ChIP-seq**. Chromatin immunoprecipitation was performed as previously described[34]. Briefly, $1 \times 10^7$ cells from primary erythroid cultures were fixed with 1% formaldehyde, before neutralization, and lysis in SDS ChIP lysis buffer (Millipore ChIP kit, 17-295). Pre-clearing was performed with protein agarose A beads before overnight incubation with antibody, and subsequent agarose bead precipitation. Washes were performed according to the manufacturer's protocol. Antibodies used were anti-pan-H4 acetylation and anti-SCL. ChIP-seq libraries were prepared and sequenced using the standard Illumina paired-end protocol.

**Antibodies used for ChIP**. Antibodies to the following proteins were used: anti-pan-H4 acetylation (Upstate 17-630; dilution 1:200) and anti-SCL (kindly supplied by Professor Catherine Porcher, MRC Molecular Haematology Unit, Weatherall Institute of Molecular Medicine, Oxford; dilution 1:200).

**Southern blotting**. Southern blotting was performed using standard methods. A 538 bp probe binding the region over MCS-R2 (chr16: 163380—163907 on build hg19) was generated by PCR, using primers FWD 5′-AACAAGAAAACCAGCAG GCTCC-3′ and REV 5′-TGTAAGTCCATCCAGGTGTGAGTTC-3′. This was cloned into the vector pGEM-T-easy (Promega) for confirmation by sequencing prior to excision and purification. About 50 ng of probe was end-labeled with α-P$^{32}$ using the megaprime kit (Amersham, RPN1604) and hybridized to blotted genomic DNA from patient MC, RC, and a normal control, digested using BamH1. A labeled fragment size of 19 kb is expected when hybridized with the MCS-R2 probe.

**Sequence analysis**. PCR sequencing was performed on PCR products purified with Qiaquick Gel Extraction Kit (Manufacturer Qiagen Cat No 28704) or Qiaquick PCR Purification Kit (Manufacturer Qiagen Cat No 28104) and then sequenced with the amplification primers using BigDye Terminator v3.1 Cycle Sequencing Kit (Manufacturer ThermoFisher Scientific Cat No 4337455) as per the manufacturer's instructions. The products were then cleaned with a standard ethanol precipitation and run on an AB3730 DNA Analyzer (ThermoFisher Scientific) and analyzed using DNA analysis software Sequencer 5.0.1 (Gene Codes Corporation).

**Colony forming unit assay**. Single-cell suspensions were mixed into Methocult Optimum (Manufacturer Stemcell Technologies Cat No H4034) to allow progenitor cells to expand and differentiate into myeloid or erythroid colonies as per the manufacturer's instructions. Colonies were inspected and picked manually into PBS using a light inverted microscope, EVOS XL Imaging System (Manufacturer ThermoFisher Scientific Cat No AME3300).

**Animal models and xenotransplantation assay**. Female NSG (NOD.Cg-Prkdc$^{scid}$ Il2rg$^{tm1Wjl}$/SzJ, Jackson Laboratory, USA) mice (species-Mus musculus) were used for xenotransplantation assays. Ten- to 14-week-old NSG mice were irradiated twice with 100 cGy, 4 h apart. Cells were administered via intra-tibial injection within 24 h of the second irradiation. Four mice were injected during the first transplantation experiment and three mice received injections during the secondary transplantation experiment. Human CD45(hCD45)+33 + 19− (myeloid) or hCD45+33-19+ (B-lymphoid) engraftment was analyzed by FACS and defined as ≥0.1% of live mononuclear cell gate. Mice were killed for cell harvesting from lower limb, pelvic, and vertebral bones at 12 weeks post injection. Permission was granted to perform animal experiments by the UK Government Home Office through the Project License 30/2465

**Statistical analysis**. Exact sample sizes and representations of error bars are indicated in each figure. All experiments were performed in triplicate unless specified otherwise. A two-tailed unpaired Student's t-test was used in statistical analysis between groups for normally distributed data and Mann–Whitney U-test was used for data, which are not normally distributed; comparison groups are

mentioned in each figure. Differences corresponding to $P < 0.05$ were considered statistically significant. No statistical method was used to predetermine sample size and data analysis was not blinded. No samples, mice, or data points were excluded from the reported analysis. No randomization method was used in experiments.

**Data availability**. All sequencing data sets are available at the National Center for Biotechnology Information Gene Expression Omnibus (NCBI GEO) with accession numbers GSE95370, GSM2508191, GSM2508192, GSM2508193, GSM2508194, GSM2508195, GSM2508196.

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

## Acknowledgements

This work was supported by grants to D.R.H. by the UK Medical Research Council (grant number MC_UU_12025/ unit programme MC_UU_12009/4) and the NIHR Oxford Biomedical Research Center. S.M. is a Commonwealth Scholar, funded by the UK government. We also acknowledge funding from Helmut Horten Foundation. We thank Anuja Premawardana of University of Kelaniya, Sri Lanka for providing samples from β-thalassemia patients and Gathsaurie Neelika Malavige of University of Sri Jayawardenapura, Sri Lanka for providing facilities for some of the tissue culture work. We thank Yale Michaels for advice using Droplet Digital PCR. Addgene plasmid 48138 (pSpCas9(BB)-2A-GFP (PX458)) was a gift from Feng Zhang. We thank family (C) for their kind collaboration. We also thank Dr João Lavinha for critically reading the paper and Dr Tudor Fulga for helpful discussion.

## Author contributions

S.M., C.A.F., D.H., L.Q., P.V., R.J.G., and D.R.H. designed the research; P.F. and A.C. provided patient data and samples; S.M., C.A.F., D.H., M.B., L.Q., K.C., P.H., D.D., J.K., M.G., J.A.S-S., J.D., B.U., P.S., J.A.S. performed the experiments; S.M., C.A.F., D.H., M.B., L.Q., J.T., J.R.H., R.J.G. and D.R.H. analyzed and interpreted the data; and S.M., C.A.F., D.H., L.Q., R.J.G., and D.R.H. wrote the manuscript.

## Additional information

**Competing interests:** The authors declare no competing financial interests.

