## [Peer Review file · Nature Communications]

REVIEWERS' COMMENTS:

Reviewer #1 (Remarks to the Author):

The authors have largely addressed my concerns.

I suggest some of the points brought up in the response to reviewers be more thoroughly integrated to the revised text such as:

1. The chance that alpha-globin expression could be brought too low, and thus the conversion of a beta-thalassemia to an alpha-thalassemia phenotype. Ways this could be improved in future iterations should be discussed. This seems the most important limitation of the approach, and not completely mitigated by the fact that rare clones with desirable expression levels may be under positive selection at the erythroid precursor level.
2. The apparent result of the xenograft experiment that just 1 clone was responsible for long-term reconstitution should be discussed explicitly.
3. The legibility of Fig 1h should be improved. The gene names are too small to read.
4. Please help reader to interpret Fig 5c, in which alpha/non-alpha globin ratio appears to be ~ 1 in beta-thalassemia, rather than expected >1 .
5. Clarify does the ddPCR design specifically detect deletions or would it also detect inversions? If also detects inversions then please clarify text/legend.
6. The off-target figure is hard to interpret, please clarify. What is y-axis? What is technical sensitivity of this assay to detect rare alleles? Also confusing why would a limited set of clones be used here rather than bulk populations in which it would be more likely for rare alleles to be present. If 30 clones tested it would seem that only a minimum of 1 in 60 off-targeted alleles could be found.

Reviewer #2 (Remarks to the Author):

Higgs and colleagues present a thoughtful and interesting advance in identifying additional genomic targets for a gene editing-based strategy to treat the hemoglobinopathies: deleting an enhancer of the alpha globin gene to reduce globin chain imbalance in beta-E thalassemia. With respect to comments made during an earlier round of review

- 1 – there are multiple efforts ongoing at increasing fetal globin (Bioverativ is using editing of the BCL11A enhancer; CRISPR and Bayer aim to delete a region in the globin locus itself; Editas has a similar effort); the manuscript refers to these efforts in the discussion and appropriately points out that the approach it describes can be used on its own, or as a complement to those efforts
- 2 – endpoint PCR with primers that lie outside the deleted region is, in specific settings, prone to producing an overestimate of the actual deletion efficiency; the authors now present digital droplet PCR data (Fig 2d) using a deleted-region-specific primer pair set, which provide a convincing measurement of deletion efficiency
- 3 – figure 6 presents analysis of single cell derived erythroid clones (rather than a pool of cells); under conditions of limited patient material availability, the authors appropriately used the cells for the better experiment (in their own words).

All the minor points raised during earlier review have been appropriately addressed in the revision.

Responses to Reviewers' Comments

Article title: Editing an α -globin enhancer in primary human hematopoietic stem cells as a treatment for β -thalassemia

Article ID: NCOMMS-17-08247-T

Reviewer #1:

The authors have largely addressed my concerns.

I suggest some of the points brought up in the response to reviewers be more thoroughly integrated to the revised text such as:

1. The chance that alpha-globin expression could be brought too low, and thus the conversion of a beta-thalassemia to an alpha-thalassemia phenotype. Ways this could be improved in future iterations should be discussed. This seems the most important limitation of the approach, and not completely mitigated by the fact that rare clones with desirable expression levels may be under positive selection at the erythroid precursor level.

Author response: We have now included a section which more thoroughly discusses the limitation of excessive down regulation of alpha-globin and potential ways which this could be improved (Page 15).

2. The apparent result of the xenograft experiment that just 1 clone was responsible for long-term reconstitution should be discussed explicitly.

Author response: We have now included a section which explicitly discusses this result (Page 15).

3. The legibility of Fig 1h should be improved. The gene names are too small to read.

Author response: We have improved the legibility of figure 1h by increasing font size of gene names.

4. Please help reader to interpret Fig 5c, in which alpha/non-alpha globin ratio appears to be ~ 1 in beta-thalassemia, rather than expected >1 .

Author response: In figure 5c, we have normalized the $\alpha/(\beta+\gamma)$ globin mRNA ratio of individual clones to the median $\alpha/(\beta+\gamma)$ globin mRNA ratio of non-deleted HbE β -thalassemia clones. Therefore $\alpha/(\beta+\gamma)$ globin mRNA ratio appears as ~ 1 . We have included this information into the figure legend as suggested.

5. Clarify does the ddPCR design specifically detect deletions or would it also detect inversions? If also detects inversions then please clarify text/legend.

Author response: The design for the ddPCR is such that it specifically detects deletions. It could not have detected inversions.

6. The off-target figure is hard to interpret, please clarify. What is y-axis? What is technical sensitivity of this assay to detect rare alleles? Also confusing why would a limited set of clones be used here rather than bulk populations in which it would be more likely for rare alleles to be present. If 30 clones tested it would seem that only a minimum of 1 in 60 off-targeted alleles could be found.

Author response: We have modified the off-target figure and figure legend to improve clarity. Figure 3e is a composite diagram of all off-target sites tested for CRISPR pair Cr2+12. The x-axis denotes 200 bases (± 100 bases from potential off-target site) of all captured sites. The y-axis denotes counts deviating from the reference sequence which are normalized to 10,000 counts. Figure 3f which represents similar analysis for CRISPR pair Cr1+8 is now moved to supplementary figure 6 which shows results of experiment using CRISPR pair Cr1+8 to simplify interpretation.

A set of clones was chosen for the analysis which were known to be edited i.e. CRISPR/Cas9 was active in these cells. Also sequencing these in depth ensured that no amplification/sequencing/mapping artifacts were causing false positive off-target hits. The referee is correct that off-target alleles of frequency $\sim 1/60$ have been excluded and that rare alleles would not be detected by this method but it would be difficult in a bulk population to distinguish off-target hits from amplification/sequencing errors and mapping artifacts.

Reviewer #2 (substitute for original reviewer 2):

Higgs and colleagues present a thoughtful and interesting advance in identifying additional genomic targets for a gene editing-based strategy to treat the hemoglobinopathies: deleting an enhancer of the alpha globin gene to reduce globin chain imbalance in beta-E thalassemia. With respect to comments made during an earlier round of review

1 – there are multiple efforts ongoing at increasing fetal globin (Bioverativ is using editing of the BCL11A enhancer; CRISPR and Bayer aim to delete a region in the globin locus itself; Editas has a similar effort); the manuscript refers to these efforts in the discussion and appropriately points out that the approach it describes can be used on its own, or as a complement to those efforts

2 – endpoint PCR with primers that lie outside the deleted region is, in specific settings, prone to producing an overestimate of the actual deletion efficiency; the authors now present digital droplet PCR data (Fig 2d) using a deleted-region-specific primer pair set, which provide a convincing measurement of deletion efficiency

3 – figure 6 presents analysis of single cell derived erythroid clones (rather than a pool of cells); under conditions of limited patient material availability, the authors appropriately used the cells for the better experiment (in their own words).

All the minor points raised during earlier review have been appropriately addressed in the revision.

Author response: There were no comments by Reviewer #2 which needed clarification or revision.